Review article

# Environmental lead risk in the 21st century

Mengli Chen [1] ✉, Ludovica Gazze[2], Francis J. DiTraglia[3], Reshmi Das [4], Jerome Nriagu[5], Yigal Erel [6], Edward A. Boyle [7], Caroline M. Taylor[8] & Dominik Weiss [9] ✉

Lead has been central to technological development for centuries; however, its release into the environment and subsequent human exposure pose significant public health risks. The review presented here critically assesses the contemporary environmental lead risk as global lead production and use are rapidly increasing, largely driven by the rising demand for electrification. We show that environmental lead exposure persists today due to legacy contamination, ongoing coal usage, and insufficient protection of workforces during production, use, and recycling of lead-acid batteries and other lead-containing products, particularly in low- and middle- income countries. We estimate that contemporary childhood lead exposure alone leads to an annual global economic loss exceeding $3.4 trillion (2021 US dollars adjusted for purchasing power parity), with pronounced disparities between high- and low- and middle- income countries. To prevent a large-scale resurgence in lead exposure, we identify four critical areas for urgent policy intervention.

Lead is a metal that has always been central to the technological development of human societies, being for example used extensively in plumbing, construction, batteries, alloys, gasoline additives, paints, and electrical appliances. However, it is also a potent toxin with significantly adverse effects on human health[1] if taken up via inhalation or ingestion. Inadequate management of energy generation, industrial processes, and transportation has resulted in environmental lead pollution over most of human history, causing waves of mortality and morbidity. The most recent and by far the largest wave was driven by the worldwide use of leaded gasoline in cars, ships and planes[2]. Subsequent bans on leaded gasoline have reduced blood lead levels (BLLs)[3] in many countries, implying that global environmental challenges can be dealt with when there is willingness.

The reductions in BLLs, however, have created the false impression that the problems of environmental lead and subsequent human exposure have been resolved[4]. Indeed, in many low- and middle- income countries (LMICs), the initial decline in BLLs has flattened or even reversed[5], and mean BLLs remain above the current World Health Organization's (WHO) recommended intervention level[6]. Different global trends observed in BLLs have been heavily influenced by the relocation of leaded gasoline production and consumption from high-income countries to LMICs over the past two decades (Fig. S2), by the increased use of coal for energy production and by the manufacture and disposal of lead acid batteries (LAB) and other lead-containing products[6,7] (Fig. 1). While the ban on leaded gasoline came after more than nine million metric tons of lead had been emitted into the environment globally (Fig. S2), current lead production is approximately 16 million tons annually[8] (Fig. 1), by far surpassing environmental lead emissions from gasoline at any time. At this scale, even minor environmental leakage poses major risks to human and ecosystem health. Monitoring and tracing sources of lead in air, water, and soils, and quantifying the socioeconomic impacts of lead poisoning remain therefore critical scientific and societal challenges of the 21st century.

In this paper, we critically assess the environmental lead risk in the 21st century. To this end, we first discuss the main lessons drawn from past studies assessing the health and environmental impacts of historical lead exposure, notably from leaded gasoline, banned globally in 2021. We then assess the current scale of global lead production, ongoing environmental sources and exposure pathways, and their socioeconomic consequences, before outlining critical gaps in knowledge and policy interventions needed to prevent renewed global harm.

[1]Tropical Marine Science Institute, National University of Singapore, Singapore, Singapore. [2]Department of Economics, University of Warwick, Coventry, UK. [3]Department of Economics, University of Oxford, Oxford, UK. [4]School of Environmental Studies, Jadavpur University, Kolkata, India. [5]School of Public Health, University of Michigan, Ann Arbor, MI, USA. [6]The Fredy and Nadine Herrmann Institute of Earth Sciences, Hebrew University of Jerusalem, Jerusalem, Israel. [7]Earth, Atmospheric and Planetary Science, Massachusetts Institute of Technology, Cambridge, MA, USA. [8]Centre for Academic Child Health, Bristol Medical School, University of Bristol, Bristol, UK. [9]Earth Science and Engineering, Imperial College London, London, UK. ✉e-mail: mengli.chen@nus.edu.sg; d.weiss@imperial.ac.uk

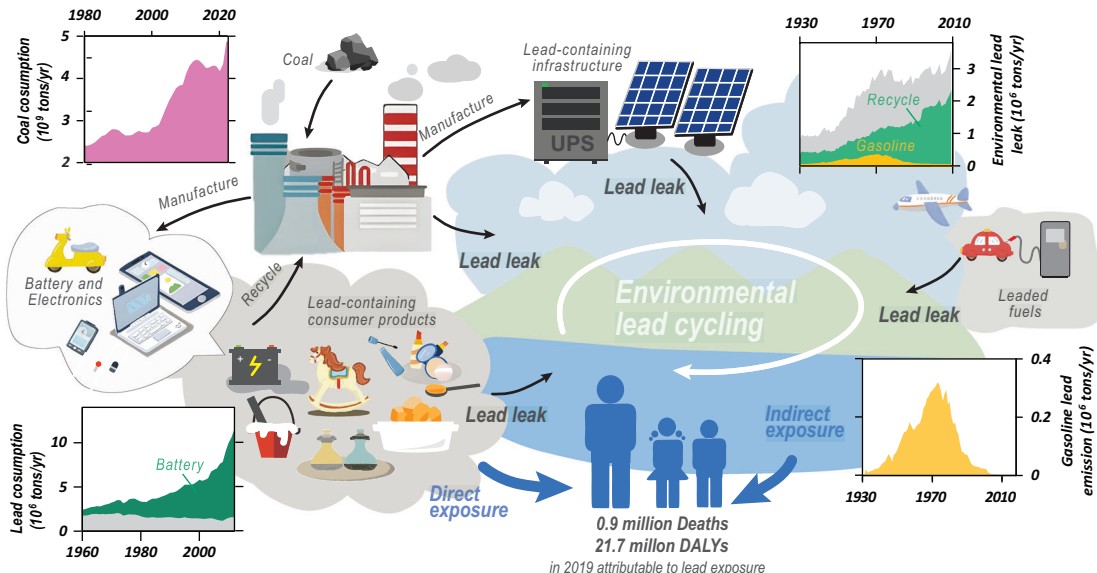

**Fig. 1 | Schematic of the environmental lead cycle.** Major transfer of lead from the anthropogenic to the natural environment during coal combustion and lead acid batteries recycling as well as the geochemical cycling of legacy lead and the consumption of lead-containing customer goods result in ongoing human lead exposure. Charts included in the figure are showing: (i) lead consumption, including coal (pink)[114] and lead acid batteries (dark green)[8]; (ii) lead leakage, including recycling of lead-containing products (largely batteries, light green)[46]; and (iii) lead emissions, including leaded gasoline (yellow; original data in Table S1) are inserted.

## Lessons learned from historical environmental lead exposure

### Major lead exposure episodes in human history

For millennia, humans have mined lead and manufactured products using it[9]. These activities have continuously released lead into the environment, leading to uninterrupted human exposure[10]. Three historical episodes of widespread lead exposure, however, stand out for their significant impact on human health.

The first arose from mining and use of lead during the Roman Empire[11], culminating with the Roman aristocracy's fondness for "sugar of lead" as an additive in food and beverages, which has been associated with saturnine gout[12]. The second took place in the 17th and 18th centuries, when Western European societies faced "Devonshire colic" stemming from the consumption of alcoholic drinks contaminated with lead[13]. The third and by far the most profound exposure arose from the use of tetraethyl lead as an anti-knock agent in automobile engines beginning in early 1920s[14] (Text S1). Lead from the exhaust of hundreds of millions of automobiles was emitted into the atmosphere, dispersing everywhere on earth.

This extensive release of lead has resulted in significant morbidity and mortality, underscoring the harmful effects of anthropogenic lead pollution on human health via environmental lead exposure[15]. At its peak, the use of leaded gasoline increased lead concentrations in aerosols by 100 to 2600 times the expected natural concentration[16], which then led to increased human exposure as indicated by elevated BLLs. Early work in the USA established the direct link between aerosol lead and blood lead (Fig. S1, Text S1) that was subsequently confirmed worldwide[3]. The introduction of catalytic converters required the removal of tetraethyl Pb[17] and this directly contributed to the phasing out of leaded gasoline, with Algeria being the final country to do so in 2021 (Fig. S2). The successful ban of leaded gasoline has been a long-overdue key milestone for global health and marks arguably one of the major policy accomplishments of the last 50 years[18].

### Impact on the environment: lead in the marine, terrestrial, and atmospheric environment

Extensive leaded gasoline usage resulted in the worldwide contamination of the natural environment with lead and the interruption of the geochemical lead cycle (Text S2, Fig. S3).

Elevated lead concentrations measured in the surface layers of Greenland ice cores and surface waters of remote seawaters led early to the conclusion that lead derived from gasoline emissions had reached every corner of the planet by 1963[19]. At the peak of leaded gasoline emissions in the 1970s, North Atlantic surface water downwind of the USA had the world's highest dissolved lead concentration, with levels >150 pmol/kg[20] (Fig. S3). These high concentrations decreased to between 20 and 40 pmol/kg following the phasing out of leaded gasoline, with naturally-sourced lead detected in some areas of the North Atlantic after the 2010s[21–24] (Fig. S3). Today, according to data reported from GEOTRACES[25], a program assessing trace element cycling in the oceans, the highest lead concentrations in open-ocean seawater are found in the surface waters of the north-western Pacific (60–100 pmol/kg)[26,27] and northern Indian Ocean (60–80 pmol/kg)[28,29], coinciding geographically with contemporary lead emissions stemming from China and India, respectively.

In terrestrial environments, enrichment of lead has been observed in the top sections of urban and rural soils around the world (Fig. S4), suggesting the residence time of lead in the terrestrial environment is significantly longer than in the marine or atmospheric environments (Fig. S3, Text S2). This strong surface enrichment makes soils an enormous repository for legacy lead (Fig. S3, Fig. S4), which will likely persist over centuries and exchange with the atmosphere and the biota. Indeed, today's aerosol lead concentrations are still hundreds of times higher than the anticipated natural baseline derived from ice core records (Fig. S5) even though measurements in the world's major cities show that atmospheric lead concentrations have stabilized over the recent decade to as low as 10 to 50 ng/m$^3$ compared with above 1000 ng/m$^3$ during the leaded gasoline era (Fig. S6). Long-range atmospheric transport leads to significant transboundary lead pollution that can be traced quantitatively using lead isotopes (Text S2). Overall, lead concentrations in the atmosphere vary significantly with location and time, reflecting variable sources, fluxes, and processes. This leads thus to human exposures that are both temporally variable and geographically uneven.

### Impact on human health: lead in blood

Human exposure to lead arises largely through ingestion of contaminated food and water, and through inhalation of lead-containing aerosols or mineral dust. Additional exposure can occur via incidental ingestion of

contaminated soil or via contact with lead-containing consumer products (Fig. 1). BLL is a common measure of lead exposure. While medical evidence suggests that no level of lead can be deemed safe[1], the current operationally defined criterion for intervention in children is a BLL of 3.5 µg/dL in the USA[30] and 5 µg/dL following WHO guidelines[31], both notably higher than the estimated pre-industrial BLL of 0.016 µg/dL[32].

During the period when leaded gasoline was widely used as fuel source, elevated BLLs were found worldwide as large amounts of soluble lead compounds, such as lead chloride or bromide, were emitted from tailpipes into the atmosphere and subsequently inhaled by humans (Text S1). Between 1972 and 1975, screening programs in the USA showed that 11% of children had BLLs above 40 µg/dL while 2% had BLLs above 80 µg/dL[33]. Results from another 27 cities not involved in these screening programs found that about 17% of children under 6 years old had BLLs above 40 µg/dL[34]. A subsequent survey conducted between 1976 and 1980 found that nearly 80% of the children 6 months to 5 years old had BLLs above 10 µg/dL, and virtually every child (99.8%) had a BLL above 5 µg/dL[35,36]. In 1984, the number of children in the USA with BLLs above 10 µg/dL was estimated to be 6.4 million, and in many urban areas, every child had a BLL above 5 µg/dL[37].

Comparable prevalence values were found in other parts of the world. Surveys conducted in the European Economic Community in 1979 and 1981 found mean BLLs above 10 µg/dL in children and adults[38,39]. Only few populations in Europe had BLLs below 5 µg/dL before the phase out of leaded gasoline[40]. Studies in urban areas of Asia, Africa, South America, and New Zealand found that BLLs in large numbers of children exceeded 5 µg/dL before the ban on leaded gasoline[40].

The inescapable conclusion is that during the 1970s and 1980s, at least 50% of the world's population had levels of blood lead that would trigger action today[41]. Comparing this value to the scale of other contemporary issues illustrates the magnitude of the lead exposure problem: approximately 35% of the world's population is forced into poverty by natural disasters[42], while about 10% of world's population was infected by COVID-19 as of June 2024[43]. Thus, the scale of environmental lead exposure from leaded gasoline was huge when the use of leaded gasoline peaked.

## Contemporary lead sources and exposure pathways

Today, ~85% of global lead production is used in the manufacture of lead–acid batteries[44] (Fig. 1), with additional applications in electrical appliances and photovoltaic cells that support telecommunications and critical uninterruptible power supplies[44]. These lead-containing products are increasingly recycled[44] and managing the recycling process sustainably is key to prevent lead leakage at the workplace into the local environment and into nearby communities[7,45]. Inadequate and illegal reprocessing practices, however, are of great concern and particularly prevalent in LMICs[46]. For example, for China alone, it has been estimated that 30% to 40% of LABs are recycled illegally[47]. In Bangladesh, Pure Earth, a non-governmental organization working on identifying, cleaning up, and preventing pollution in LMICs, identified more than 300 informal LAB recycling sites and expect many more that remain undocumented[45]. Other cases of lead poisoning linked to inadequate recycling over the past decade have also been reported in Senegal[48], India[49], and Vietnam[50]. The total population affected by an estimated 10,000–30,000 informal LAB processing sites in LMICs is likely to range from 6 to 16.8 million[51]. The international transport of lead-containing waste from developed countries (outgoing) to developing countries (incoming) may disproportionally increase lead exposure in disadvantaged communities. For example, India, where 25% to 40% of LABs were recycled illegally in 2023, received approximately 0.4 million metric tons of lead waste between 2018 and 2022[52], which accounted for 15% of the global lead waste trade. Used LABs are also shipped from the USA to Mexico for recycling[53].

Coal combustion remains a significant source of lead in the environment due to high concentrations of lead in impure coal, large-scale coal usage for energy production, and lack of emission controls, especially in LMICs[54]. Coal combustion contributed approximately 50% of the atmospheric lead emissions in China in 2009[55], 12% to 42% in India in the 2010s[56], and about 30% globally in 2005–2012[57]. Implementation of adequate emission control technologies, such as flue gas desulfurization systems, however, reduce this lead source without decreasing coal consumption. This has been demonstrated recently in China: lead emissions from coal-fired powerplants decreased from about 1200 metric tons in 2005 to about 440 metric tons in 2020[58].

Continuous elevated lead levels compared to anticipated natural baseline in aerosols around the world's major cities with limited contemporary emission (Fig. S6) suggest that as discussed above remobilization of surface polluted soil is an important source of lead in the atmosphere[59] (Fig. S3). This remobilization of legacy lead suggests ongoing, possibly permanent, exposure of the population mediated by natural processes like dust resuspension. Mass balance calculations and isotope signatures show that legacy gasoline lead contributes today about 30% to 40% of London's aerosols[60] and about 10% to 40% of Singapore's aerosols[56]. In contrast, in countries with lower historical emissions and/or higher contemporary emissions, the contribution of gasoline legacy lead to the atmosphere is smaller, for example 4% to 9% in India and 4% to 20% in Vietnam[56]. Lead-containing paint, soldering products, spent ammunition from firearms, and lead pipes, are other important legacy pollution sources leading to lead exposure. For example, leaded paint, banned for several decades in the USA (Fig. S1), is still the main source of lead exposure in many older homes as painted surfaces release lead-containing dust over decades and centuries[61], and the Flint water crisis[62] drew recently attention to the risks of lead leaching from aging water infrastructure. The remobilization of legacy lead underscores the enduring impact of historical emissions and constructions and provides an important warning that contemporary emissions could become legacy sources in the future.

Finally, particularly in LMICs, increasing lead exposure also arises from the consumption and use of customer products that contain lead, sometimes fraudulently added. These include leaded paint, soldering, cosmetics, glazed ceramics, toys[7], spices[63], and traditional medicines[64] (Fig. 1). For example, studies in Bangladesh found adulteration of turmeric, an essential ingredient in South Asian daily cooking, with lead chromate for coloring, resulting in lead concentrations to up to 500 times the national standard[63]. Subsequent nationwide efforts removed contaminated turmeric from the market, highlighting the effectiveness of targeted policy interventions, including enhanced food testing and public education[65]. The continued use of lead-based paint presents a similar concern. It remains legal and is still sold in many Asian, African, Latin American, and European countries (about 50% of countries worldwide), and lead concentration in paint can exceed 10,000 mg/kg[66].

As the lead in the environment originates from increasingly diverse sources, each of which is releasing lead in distinct chemical forms with contrasting solubilities, exposure pathways today differ with location and site specific monitoring and management strategies are required[67,68]. For example, oral ingestion is often the dominant route of exposure near recycling facilities and coal-fired power plants, likely due to the consumption of food contaminated with atmospheric particulates. Food has been identified as a primary pathway in urban centers of India[69], Bangladesh[70], Brazil[71], Nigeria[72], and Vietnam[73]. Inhalation and ingestion are important in regions affected by diffuse lead sources[74].

## Socio-economic impact of environmental lead pollution

Existing epidemiological and economic evidence shows that lead exposure affects human health and childhood development with important economic consequences.

### Present evidence for important socioeconomic and public health global lead burden

The 2019 Global Burden of Disease Study (GBD) estimated that approximately 0.9 million deaths and 22 million lost disability-adjusted life years are attributable to lead exposure each year[5] (Fig. 1). Lead is estimated to account

for nearly half of the 2 million lives lost to chemical exposure each year[75], almost one-third of the global burden of idiopathic intellectual disability, which includes delayed or impaired speech, language, motor condition, and visuo-spatial skills, and 4.6% and 3% of the global burden of cardiovascular disease and chronic kidney disease, respectively[75]. Oehlsen (2024)[76] concluded that lead exposure remains an important but neglected problem globally, noting that deaths from lead exposure "fall between HIV/AIDS (approximately 864,000 annual deaths) and tuberculosis (approximately 1.18 million annual deaths)", while "our best guess of philanthropic spending in reducing lead exposure per disability-adjusted life-year lost is 45 cents, which is two orders of magnitude lower than our estimate of the equivalent figure for malaria and tuberculosis, themselves neglected diseases"[76]. Here, we review what is currently known about the effects of lead exposure before discussing methodological issues and data gaps that highlight the need for further research.

The large global burden of lead is due to the extensive reach it has in the human body even in small concentrations due to its ability to replace calcium, allowing it to affect a wide range of organ systems. Lead affects the central and peripheral nervous system[77], the hematopoietic system, the renal system, the reproductive system, and the cardiovascular system, causing encephalopathy, peripheral neuropathy, renal dysfunction, hypertension and cardiovascular disease, infertility, miscarriage, and pre-eclampsia[78]. Moreover, after circulating in the bloodstream for an estimated 38 days following ingestion or inhalation, lead is stored in bone tissue and can be released later in life when the body loses bone mineral density or has an increased demand for calcium, e.g. during pregnancy[1]. At low BLLs, lead exposure is generally asymptomatic at first, with permanent damage potentially manifesting years later[1,79]. High BLLs can lead to immediate and acute symptoms such as seizures and gastrointestinal issues[1].

While lead exposure has negative consequences at all ages, it is particularly harmful for the developing fetus and children. Lead crosses both the placenta and the blood–brain barrier, making it especially damaging to the developing central nervous systems of the fetus and young children[80]. Experimental studies in animal models show that the fetus is particularly vulnerable to the effects of lead because lead affects processes critical to the development of the central nervous system[81,82]. Thus, relatively low levels of exposure that do not greatly harm the mother may adversely affect fetal development and therefore affect subsequent development and behavior during childhood[81,83]. This can occur through direct toxicological effects or epigenetic effects[84,85]. Children have an enhanced exposure to lead due to hand-to-mouth transfer of dust and soil, and also absorb a greater proportion of the lead they ingest than adults[80]. Childhood lead exposure has been associated with reduced IQ[86], lower cognitive function[87], impaired academic performance and behavioral problems[88], increased risk of attention deficit hyperactivity disorder[89], and hearing loss[90]. Importantly, all these physiological pathways likely affect children's cognitive abilities, as evidenced by a large literature showing that lead exposure is associated with worse educational outcomes globally[91–95].

Due to lead's negative impacts on impulse control and aggression, low-level early childhood lead exposure increases youth behavioral issues and crime[95,96]. Aizer and Currie estimated that 75% of the decline in school suspensions between 1994 and 2015 observed in Rhode Island could be explained by a reduction in BLLs following the ban on leaded gasoline and remediation of lead paint hazards in homes[95]. However, the effects of early childhood lead exposure can persist into adulthood[97,98]. For example, Grönqvist et al. found that childhood lead exposure in Sweden causes a higher probability for criminal conviction up to age 35 years, as well as lower high school graduation rates[96]. While lead exposure in the developed world has declined markedly in recent decades, these long-term effects remain relevant today. McFarland et al. found that about half of the US adult population in 2015 had childhood BLLs in excess of 5 μg/dL[99], the CDC's upper reference range value for childhood BLLs at that time. The current value has been lowered to 3.5 μg/dL[30].

BLLs above 5 μg/dL in adults have been associated with increased all-cause mortality and increased mortality from cardiovascular and ischemic heart disease[100]. The 2019 GBD estimated that 0.85 million cardiovascular disease deaths in 2019 were attributable to lead exposure[5]. Larsen and Sanchez-Triana suggested that this value could be a substantial underestimate, by as much as a factor of six[101]. Hollingsworth and Rudik found that airborne lead pollution during NASCAR races, which allowed the use of leaded gasoline until 2007, increased mortality in elderly people[102]. A review by Ramirez Ortega et al. suggested that lead exposure causes decreases in cognitive function among adults as well, including decreases in short-term memory and motor control alterations[80].

While the biological mechanisms through which lead exposure affects the human body are well understood, the evidence for the magnitude of the resulting effects on cognitive, health, and behavioral outcomes described above is based on observational data rather than data from experimental studies, due to obvious ethical constraints. Lead exposure is not randomly assigned: it is robustly associated with a wide range of socio-economic factors that likely affect the same health, educational, and behavioral outcomes[103]. For instance, lower-income neighborhoods typically feature older housing stock, with associated risks of lead exposure from paint and pipes, and are more likely to be located on or near former industrial areas[104]. Because socio-economic status is directly related to health, educational, and behavioral outcomes, as well as to lead exposure, it is challenging to separate the causal effect of lead from its association with confounding variables.

Most observational studies fall into two broad groups based on their approach to resolving this methodological challenge. The first relies on regression adjustment to account for known, observed confounding factors[89,100]. As an example, Crump et al. adjusted for birth weight, maternal IQ, maternal education, maternal alcohol, maternal tobacco usage, and birth order, among others, when estimating the dose–response relationship between lead exposure and IQ scores[105]. The second group responds to the concern raised by Dominici et al. that "associational approaches to inferring causal relations can be highly sensitive to the choice of the statistical model and set of available covariates that are used to adjust for confounding"[106]. This type of research relies on quasi-experimental (QE) variation in lead exposure "determined naturally by politics, an accident, a regulatory action, or some other action beyond the researcher's control" that "plausibly mitigates biases arising from confounding factors"[77]. Examples of this type of study include Aizer et al., Aizer and Currie, Grönqvist et al., and Hollingsworth and Rudik[92,95,96,102].

This difference in approach is important: the associational regression adjustment approach and the QE approach often produce substantially different estimates of the same causal effects. Higney et al. found systematically smaller estimates of the causal effect of lead on crime in the US among associational studies than in QE studies[107]. On the other hand, Crawford et al. noted that, among five QE studies relying on an instrumental variable approach that permits direct comparison between associational and QE estimates based on the same data, QE (instrumental variables) estimates of the effect of lead on cognitive outcomes were systematically higher[91]. This pattern may be explained by measurement error in the exposure variable. For example, BLLs are often measured by a finger prick (capillary) test, and even more accurate measurements from venous blood samples measure only recent short-term exposure, given the short half-life of lead in blood. Because the instrumental variables approach corrects for measurement error of the form that is empirically plausible in this context[92], this provides another argument for the advantages of a QE approach.

An important limitation of most of the QE studies described above is that they rely on data from higher-income countries, where data on BLLs and outcome measures are more readily available. Thus, while these studies have a high degree of internal validity, their results may not be applicable to LMICs where lead exposure remains high and widespread. Indeed, even many high-income countries lack representative recent population-level data on lead exposure (e.g., the UK). There is a continued need to improve the monitoring of lead exposure across the world. As pointed out by Clay et al.[77], however, "even in the absence of data on BLLs, quasi-experimental studies provide policy-relevant evidence on lead exposure outcomes." All

that is required, in principle, is an outcome measure and a source of quasi-random variation that is known to affect lead exposure.

## Calculation of earning losses

Population-level lead exposure remains significant in the post-leaded gasoline world, especially in LMICs[6]. This continued exposure reflects the persistence of legacy lead along with the emergence of new emission sources, and suggests a transition to chronic, potentially asymptomatic, environmental lead exposure from a multitude of sources. To help appreciate the magnitude of the continued harm from lead exposure, we here estimate the global cognitive damage from childhood lead exposure. The decision to focus on cognitive effects in children, rather than broader health and behavioral impacts across all age groups, is informed by measurement issues and data availability constraints.

First, BLLs, the most commonly available lead exposure measure, primarily reflect recent acute exposure rather than cumulative exposure. Despite this, BLLs are a better proxy for cumulative lead exposure in children than adults, because children have a shorter period of potential exposure. Second, attributing mortality and morbidity to lead exposure requires data that align past lead exposure with current health outcomes. For adults, while we have ready access to current data on mortality and morbidity, comprehensive historical lead exposure data are often lacking, especially on a global scale. Conversely, for children, we often have access to current lead exposure data but lack data on the long-term health outcomes that may manifest later in life, as well as the ability to causally link childhood exposure to later-life outcomes. Furthermore, quantifying the global costs associated with the well-documented behavioral effects of lead exposure, such as increased impulsivity and aggression, presents significant methodological challenges due to the multifactorial nature of behavior and the lack of standardized global measures. For example, one would need data on criminal justice costs for both juveniles and adults on a global scale. Lastly, we rely on established literature in economics and public health that provides a framework to translate cognitive impacts into monetary values and allows for a clearer attribution of cause and effect. By focusing on cognitive damage in children, we aim to provide a more robust and conservative estimate of lead's global impact. However, it is important to note that this approach likely underestimates the total global socioeconomic burden of lead exposure, as it does not account for long-term health effects, behavioral impacts, or cumulative exposure in adults.

While developed independently, our approach is similar to that of Larsen and Sanchez-Triana, who estimate a total monetary value of about $1.3 trillion (2019 USD) assigned to "future income losses from IQ loss" among children aged up to 5 years[101]. Our calculations, described below, yield a somewhat higher total value. This likely reflects different methodological choices along with a different choice of scaling. We describe here our procedure for computing the costs of global cognitive damage. Full methodological details appear in Text S4, and full replication code and data are available from the public repository: https://github.com/fditraglia/lead-review-maps. In summary, we proceeded in four steps. The first step uses GBD data to approximate the distribution of BLLs among 0–19-year-olds in each country in 2019. The second step converts the distribution of BLLs in each country to a mean number of IQ points lost per young person[105]. This provides a measure of the mean cognitive damage from lead exposure. The third step estimates the economic costs of this cognitive damage by combining cross-country data on the returns to education—a proxy for the relative returns to cognitive skills—with estimates of the reduction in lifetime earnings from lower cognitive ability taken from the US Environmental Protection Agency[108]. The fourth and final step conducts a sensitivity analysis by varying the assumptions, methods, and input data used in steps one and two.

The economic costs of cognitive damage from childhood lead exposure are reported in two ways. The first gives the mean foregone lifetime earnings in percentage points for a given country (relative IQ cost). A value of 5% for relative IQ cost, for example, would indicate that a sample person aged 0–19 years in a particular country will earn 5% less over their lifetime because of

lead exposure. The second converts these percentages into dollar values by multiplying them by GDP per capita at purchasing power parity in constant 2021 international dollars and by the population aged 0–19 years. Summing the result across countries gives total IQ cost.

Figure 2a, b shows the estimated percentage of children aged 0–19 years with BLLs above 5 and 10 µg/dL in 199 countries in 2019, based on GBD data. Lead exposure in LMICs notably exceeds that in high-income countries. This disparity likely arises from earlier bans on leaded gasoline in high-income countries, coupled with the natural renewal of the automotive stock, and stricter regulation of the industrial and domestic use of lead more generally. Our cost estimates are shown in Fig. 2c and Table 1: Fig. 2c shows relative IQ cost values for countries in our sample in 2019; Table 1 shows mean values of the relative cost by continent and the total (global) IQ cost measure. Table 1 shows the baseline results along with several sensitivity analyses described in detail in Text S4.

In general, relative IQ costs are higher in LMICs (Fig. 2 and Table 1). Nevertheless, economic losses are not necessarily proportional to lead exposure prevalence. This is because some countries with lower lead levels have relatively high returns to education, our proxy for the value of cognitive skills. As such, the per capita economic losses appear more uniform across countries than lead exposure itself. A caveat regarding the use of returns to schooling to construct Fig. 2c is that differences in these returns reflect, among other factors, systemic, and historical differences in environmental and political conditions. Arguably, these differences stem from the same global power imbalances that generated differential lead exposure burdens. Bearing this caveat in mind, the calculations described above nonetheless provide a reasonable lower bound on global damages from lead exposure. The resulting estimates span a wide range, from about 1.7% to 29.7%, with the median country losing about 7% of lifetime earnings population-wide. For purposes of comparison, Larrimore et al. report that median pre-tax income fell by about 4% in the US during the great recession[109] between 2007 and 2008, while Atwood (2022) found that the introduction of the measles vaccination in the US increased adult income by 1.1% through improved productivity[110]. Our estimate of the global yearly monetary loss from childhood lead exposure is approximately $3.4 trillion (constant 2021 international dollars, that is values are expressed in US dollars and adjusted for purchasing power parity), just over 2% of global GDP in 2019 (Table 1). Importantly, this value shows only returns for the current cohort of children aged 0 to 19 years (approximately one-third of world population), but reducing lead exposure will generate additional returns for older cohorts as well as future ones.

The sensitivity analyses show that relative IQ costs in LMICs are less sensitive to methodological choices than those in high-income countries in Europe, North America, and Oceania (Table 1). This is because high-income countries have lower mean lead exposure and the alternative assumptions that we explore in our sensitivity analysis are more consequential at the lower range of BLLs. Across different sensitivity analyses, the total IQ cost ranges from $780 billion to $4.9 trillion (constant 2021 international dollars) per year. While the precise value depends on modeling assumptions, the overall order of magnitude is stable, suggesting that the human and economic costs of unaddressed lead contamination remain significant worldwide.

## Lessons learnt and outstanding questions

Environmental lead remains a complex global challenge and continued efforts to curb exposure and its socioeconomic costs are essential to safeguard future generations. We identify to this end four urgent priorities for action.

First, minimize leakage of lead across the life cycles of lead-containing products. Rising demand for solar panels, lead–acid batteries, and other electronic materials linked to the transition to net-zero emissions risks amplifying environmental releases. Rigorous life-cycle analyses and the adoption of best practices are essential to ensure their sustainable use.

Second, reduce environmental exposure, particularly in low- and middle-income countries. Unsafe recycling of lead–acid batteries and

a. *Fraction of children with BLL > 5 micrograms/deciliter*

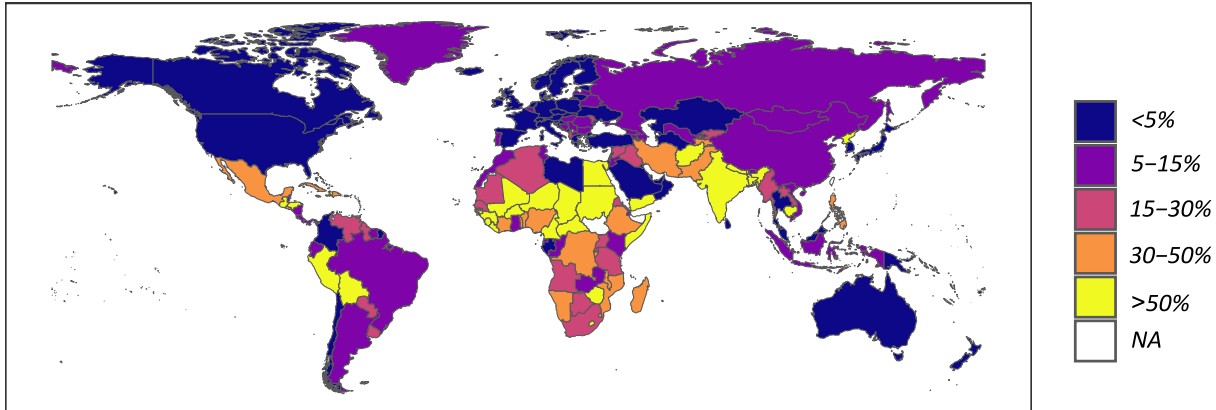

b. *Fraction of children with BLL > 10 micrograms/deciliter*

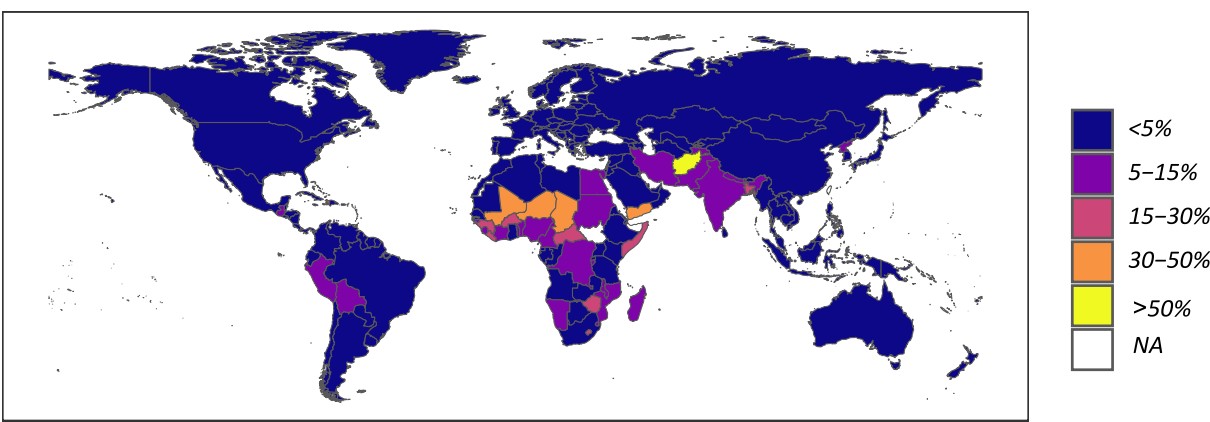

c. *Relative IQ cost of lead exposure*

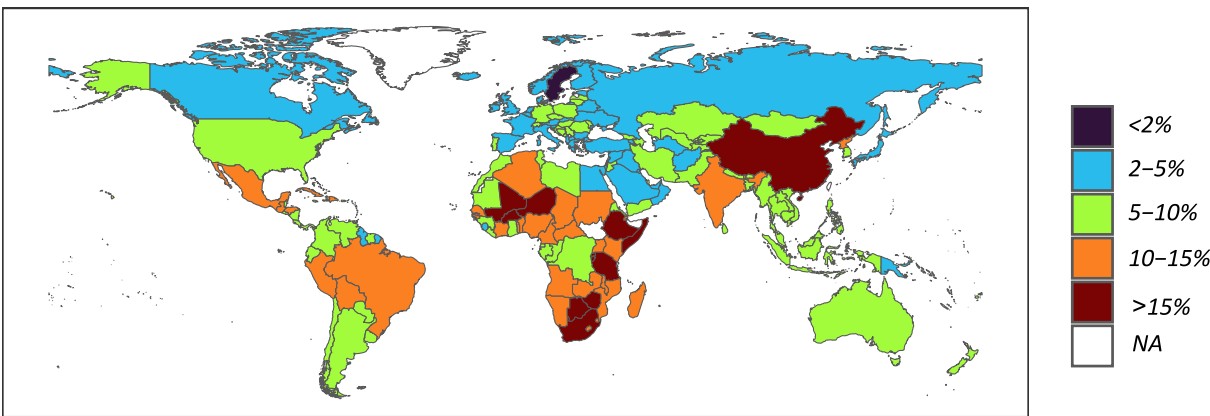

**Fig. 2 | Estimated lead exposure and IQ costs of lead exposure across countries.** The figure shows estimated percentage of children aged 0–19 years in 2019 with elevated blood lead levels: **a** above 5 µg/dL (level of concern in USA between 2012 and 2021); and **b** above 10 µg/dL, using data from the Global Burden of Disease[5]. Panel **c** shows the estimated mean percent of lifetime earnings lost per child aged 0–9 years due to cognitive losses associated with lead exposure in each country. See Text S4 for details on the derivation of the underlying data.

adulteration of foods, paints, and ceramics expose both workers and consumers to high risks. Targeted interventions — strengthened occupational protections, improved product standards, and enforcement against illegal practices — are critical to mitigate these impacts.

Third, expand environmental monitoring and improve source attribution capabilities. Early detection of lead leakage is essential for effective intervention, yet large-scale monitoring across populations, consumer goods, and environmental reservoirs is costly and logistically and politically challenging[111]. Advances in remote sensing, machine learning[112], and low-cost sensors for lead measurements in environmental matrices and blood offer new opportunities, particularly for low- and middle-income countries[101]. Integrating local knowledge and involving at-risk communities will be critical to identify hotspots and ensure interventions are effective and equitable.

**Table 1 | Estimated global cognitive damage from childhood lead exposure**

| | Baseline[a] | log-normal[b] | Lower bound[b] | Lower CI[b] | Upper CI[b] | 5% increase[b] | 10% increase[b] |
|---|---|---|---|---|---|---|---|
| **Relative IQ cost (expressed in percentage of earning loss,%)** | | | | | | | |
| Africa | 12.3 | 11.9 | 5.3 | 6.9 | 17.7 | 12.7 | 13.0 |
| Asia | 10.7 | 7.5 | 3.8 | 6.1 | 15.4 | 10.9 | 11.2 |
| Europe | 4.2 | 1.2 | 0.4 | 2.4 | 6.1 | 4.5 | 4.6 |
| North America | 5.2 | 0.7 | 0.2 | 2.9 | 7.5 | 5.5 | 5.7 |
| Oceania | 5.5 | 1.0 | 0.3 | 3.1 | 7.9 | 5.7 | 5.9 |
| South/Central America | 10.2 | 6.9 | 2.9 | 5.7 | 14.6 | 10.2 | 10.4 |
| **Total IQ cost (expressed in 2021 international dollars)** | | | | | | | |
| Global | $3.4 T | $1.8 T | $780B | $1.9 T | $4.9 T | $3.5 T | $3.6 T |

*T* trillion, *B* billion, *CI* confidence interval.
[a]Main estimates.
[b]Sensitivity analysis.

Fourth, improve our understanding of the social and economic consequences of low-level lead exposure. Exposure disproportionately affects socioeconomically disadvantaged populations, with impacts shaped by children's social environments[113]. Advancing this understanding requires accurate models[101], granular population-level blood lead data, and interdisciplinary collaboration across environmental science, public health, and social sciences. Engaging local communities in research is essential to link environmental sources to health outcomes and to design context-sensitive interventions.

## Data availability

The full replication materials for the economic calculations are available from the GitHub repository for this paper at https://github.com/fditraglia/lead-review-maps. The editable format for gasoline lead emission data, and the aerosol lead concentration data in selected cities (Fig. S6) are available on https://doi.org/10.25540/HXFN-68FM.

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

## Acknowledgements

Thanks go to Dr. Alex Hollingsworth, Dr. Heather Klemick, Dr. Claudia Persico, Ms. Iravaty Ray, Dr. Ning Zhao for their help in data mining and/or generous feedback on the manuscript. This study is supported partly by OCE-2148916 from the US NSF. M.C. and D.W. acknowledge the support from ICL-NTU funding (INCF-2023-015) in facilitating discussions. C.M.T. is funded by an MRC Career Development Fellowship (grant no. MR/T010010/1). Special thanks to the anonymous reviewers. We thank designers from Wuhan 147 Network Technology Co., Ltd. for creating part of the illustrations. We finally thank Angela L. Bandemehr and Valerie Zartarian from the USEPA for helpful discussions.

## Author contributions

Mengli Chen: Conceptualizing, Funding acquisition, Project Administration, Data mining, Illustration, Formal Analysis, Writing-original draft, Writing-review and editing. Ludovica Gazze: Conceptualizing, Data mining, Illustration, Formal Analysis, Writing-original draft, Writing-review and editing. Francis J. DiTraglia: Conceptualizing, Data mining, Illustration, Formal Analysis, Writing-original draft, Writing-review and editing. Reshmi Das: Data mining, Illustration, Writing-original draft, Writing-review and editing. Jerome Nriagu: Conceptualizing, Writing-original draft, Writing-review and editing. Yigal Erel: Writing-original draft, Writing-review and editing. Edward A. Boyle: Funding acquisition, Writing-original draft, Writing-review and editing. Caroline M. Taylor: Funding acquisition, Writing-original draft, Writing-review and editing. Dominik Weiss: Conceptualizing, Funding acquisition, Project Administration, Data mining, Illustration, Formal Analysis, Writing-original draft, Writing-review and editing.

## Competing interests

The authors declare no competing interests.
