## [Transparent Peer Review file · Communications Earth & Environment]

Environmental Lead in the 21st Century

Corresponding Author: Professor Dominik Weiss

Version 0:

Decision Letter:

Dear Professor Weiss,

Your manuscript titled "Environmental Lead in the 21st Century" has now been seen by our reviewers, whose comments appear below. In light of their advice we are delighted to say that we are happy, in principle, to publish a suitably revised version in Communications Earth & Environment.

We therefore invite you to revise your paper one last time to address the remaining concerns of our reviewers. At the same time we ask that you edit your manuscript to comply with our format requirements and to maximise the accessibility and therefore the impact of your work.

EDITORIAL REQUESTS:

*****Please take care to match our formatting and policy requirements. We will check revised manuscript and return manuscripts that do not comply. Such requests will lead to delays. *****

SUBMISSION INFORMATION:

OPEN ACCESS:

Communications Earth & Environment is a fully open access journal. Articles are made freely accessible on publication. For further information about article processing charges, open access funding, and advice and support from Nature Research, please visit <https://www.nature.com/commsenv/open-access>

Link Redacted

**** This url links to your confidential home page and associated information about manuscripts you may have submitted or be**

reviewing for us. If you wish to forward this email to co-authors, please delete the link to your homepage first **

Best regards,

Somaparna Ghosh, PhD
Associate Editor,
Communications Earth & Environment
Consulting Editor,
Communications Sustainability

REVIEWERS' COMMENTS:

Reviewer #1 (Remarks to the Author):

As a re-review of a review paper I previously reviewed for Nature. I am pleased to see the authors have taken on board my comments and comments of other reviewers. Overall, I think the paper is a valuable contribution and worthy of publishing. It is a little disappointing to see that there are still a number of sloppy typos and the language needs attention in a range of places for clarity of understanding and to remove colloquial language.

A couple of more major comemnts:

Title: I don't think the title really captures the scope of the paper, and is very general. At the very least should have the world pollution in. This seems to be an issue throughout. Environmental lead should be changed throughout to something such as 'environmental lead contamination' or 'environmental lead pollution. The title should capture the nature of the review and the assessment on earnings etc.

Abstract: The abstract seems in need of some attention (detailed comments below). Also, it should more clearly describe in say Line 36 the new aspects of this study in terms of the methodology and what it adds.

Figures: The scale bar color schemes in Fig 2 is terribly unclear and needs changing. They do not all show 0, nor do they all show 100%.

Minor Comments:

The abstract seems in need of attention. For example:

- L26 'it's industrial cycle' seems rather unclear
- L27 'critically' should be 'critical'
- L27 'paper' should be 'study' or similar
- L30-32 Isn't it already well known that this has been a big pollution event?
- L33-34 'needs our attention' is rather colloquial and inspecific
- L34 The meaning of 'this continuance' is not very clear
- L36 'we prove' should be 'we show'
- L37 'current exposure' is a vague statement and should be made clearer
- L39 what does '2021 international dollars' mean?
- L42 what does inadequate mean here?
- L44 and 47 Changing to 'lead-containing' might help clarity
- L45 lead hotspots?
- L49 Correlated to, or driven by?
- L56 Change 'likewise' to 'also'
- L57 'extensive reach' is not a clear term
- L65 Should say problem of 'environmental lead pollution' or similar.
- L83 Do you mean lead production, or release of lead into the environment? This seems very unclear.
- L120 Too colloquial - globes do not have corners. What about remote Antarctica?
- L134 'Atmosphere' and 'biota' are not 'environments'. I would reword
- L142 I would re-word "Overall, lead in the environment changes over space and time", since this applies to everything. Something such as 'Lead concentrations vary significantly with location and timing'
- L147 what does intake of aerosols or soils mean? Why is inhalation indirect?
- L150 Surely some level of lead is safe. It will never be 0. Perhaps 'detectable level'?
- L155 How is atmospheric lead ingested?
- L172-3 This sentences is very colloquial and not scientific. Remove sentence.
- L177 I think this needs to be re-worded. You reviewed findings, you didn't present new demonstration of these things. It should say after 'reviewing'
- L205. lead does not need to be capitalized.
- L218 do you mean firearms?

L219 Lead levels in drinking water have been an acute crisis in the USA in recent years – maybe could be made more of here?

L232 Remove 'indeed'

L235 'Another example is lead paint is a sentence fragment, and should be re-worded.

L242 This sentence is poorly worded and should be changed. Change 'is consequently resulting in' to 'results in an increasingly...'

L243 'This' is an inspecific way to begin, you could connect to previous sentence with a ;

L246 How are people ingesting near to lead sources? Needs explanation.

L293 I am not sure 'outcomes' is the right word here. Also, wouldn't the effects you go on to list actually be cognitive.

L326 Should this be 'confounding factors' or similar?

Reviewer #2 (Remarks to the Author):

The authors have responded very effectively to the concerns raised in the previous reviews of this manuscript. The review is much more focused and better organized, and the writing is much tighter throughout.

Reviewer #1

As a re-review of a review paper I previously reviewed for Nature. I am pleased to see the authors have taken on board my comments and comments of other reviewers. Overall, I think the paper is a valuable contribution and worthy of publishing. It is a little disappointing to see that there are still a number of sloppy typos and the language needs attention in a range of places for clarity of understanding and to remove colloquial language.

Reply:

We are grateful for the positive feedback and the time and effort dedicated to reviewing our manuscript. We hope we have addressed the language issues highlighted, and the revised version has been proofread by several native English-speaking co-authors.

Title: I don't think the title really captures the scope of the paper, and is very general. At the very least should have the world pollution in. This seems to be an issue throughout. Environmental lead should be changed throughout to something such as 'environmental lead contamination' or 'environmental lead pollution. The title should capture the nature of the review and the assessment on earnings etc.

Reply:

We have revised the title into "Environmental Lead Risk in the 21st Century" to capture the exposure to human and assessments on earnings. We also changed to environmental lead contamination and/or pollution where we felt it was appropriate.

Abstract: The abstract seems in need of some attention (detailed comments below). Also, it should more clearly describe in say Line 36 the new aspects of this study in terms of the methodology and what it adds.

Reply:

We have addressed the issues in the abstract following the detailed comments below by the referee. Again we appreciate her/his/their efforts very much. We highlight now the new aspects of the study.

Figures: The scale bar color schemes in Fig 2 is terribly unclear and needs changing. They do not all show 0, nor do they all show 100%.

Reply:

We have revised Figure 2 to make it clearer.

The abstract seems in need of attention. For example:

L26 'it's industrial cycle' seems rather unclear

Reply:

We removed 'industrial cycle'.

L27 'critically' should be 'critical'

Reply:

We now reworded to 'this study assess...'

L27 'paper' should be 'study' or similar

Reply:

We changed the word to 'study' as suggested

L30-32 Isn't it already well known that this has been a big pollution event?

Reply:

This sentence was removed to comply with the abstract's word count requirement.

L33-34 'needs our attention' is rather colloquial and inspecific

Reply:

We have removed it

L34 The meaning of 'this continuance' is not very clear

Reply:

We have reworded as: "...the environmental lead exposure persists due to..."

L36 'we prove' should be 'we show'

Reply:

We reworded the sentence and removed 'we prove'. The new sentence reads: 'Current exposure harms human health and carries considerable economic costs.'

L37 'current exposure' is a vague statement and should be made clearer

Reply:

This sentence was removed to comply with the abstract's word count requirement.

L39 what does '2021 international dollars' mean?

Reply:

This is the terminology used by the World Bank, our ultimate data source. It means the monetary values are expressed in US dollars and adjusted for purchasing power parity (PPP) based on year 2021 so that figures for all countries are comparable.

L42 what does inadequate mean here?

Reply:

We reworded to 'insufficient protection during production, use, and recycling of lead-acid battery.'

L44 and 47 Changing to 'lead-containing' might help clarity

Reply:

We agree and changed.

L45 lead hotspots?

Reply:

This sentence was removed to comply with the abstract's word count requirement.

L49 Correlated to, or driven by?

Reply:

This sentence was removed to comply with the abstract's word count requirement.

L56 Change 'likewise' to 'also'

Reply:

This sentence was removed to comply with the abstract's word count requirement.

L57 'extensive reach' is not a clear term

Reply:

We simplified the sentence to 'Lead is also a potent toxin with significant adverse effects on human health'.

L65 Should say problem of 'environmental lead pollution' or similar.

Reply:

We changed the wording as advised

L83 Do you mean lead production, or release of lead into the environment? This seems very unclear.

Reply:

We refer here to lead production, not lead release into the environment. We added now the following sentence for clarity:

At this scale, even a small leakage may cause a large enough exposure to humans and the environment.

L120 Too colloquial - globes do not have corners. What about remote Antarctica?

Reply:

We changed from 'every corner of the globe' to 'everywhere on earth'.

L134 'Atmosphere' and 'biota' are not 'environments'. I would reword

Reply:

Reworded to “Additional exposure can occur via incidental ingestion of contaminated soil or contact with lead-containing consumer products”.

L142 I would re-word “Overall, lead in the environment changes over space and time”, since this applies to everything. Something such as ‘Lead concentrations vary significantly with location and timing’

Reply:

We have changed it as proposed

L147 what does intake of aerosols or soils mean? Why is inhalation indirect?

Reply:

This is correct, we tried to squeeze too much information into the original sentence, and now we reworded the sentence to:

‘Human exposure to lead arises through ingestion of contaminated food and water, and through inhalation of lead-containing aerosols or dust. Additional exposure can occur via incidental ingestion of contaminated soil or contact with lead-containing consumer products.’

L150 Surely some level of lead is safe. It will never be 0. Perhaps ‘detectable level’?

Reply:

We have reworded this sentence

L155 How is atmospheric lead ingested?

Reply:

We removed ‘ingestion’.

Reworded to ‘ingestion of contaminated food and water, and through inhalation of lead-containing aerosols or mineral dust’.

L172-3 This sentences is very colloquial and not scientific. Remove sentence.

Reply: We think that, contextualizing the lead problem alongside today’s major health issues helps improve the accessibility and relevance of the writing for a broader audience. We think as such the sentences are valid and would like to keep them.

L177 I think this needs to be re-worded. You reviewed findings, you didn’t present new demonstration of these things. It should say after ‘reviewing’

Reply: Changed.

L205. lead does not need to be capitalized.

Reply:

This has been changed.

L218 do you mean firearms?

Reply:

This has been changed.

L219 Lead levels in drinking water have been an acute crisis in the USA in recent years – maybe could be made more of here?

Reply:

We added this information. We write now 'Similarly, the leaching of lead from aging water infrastructure—most notably highlighted by the Flint water crisis—drew attention to the risks posed by neglected infrastructure'.

L232 Remove 'indeed'

Reply:

This has been removed.

L235 'Another example is lead paint is a sentence fragment, and should be reworded.

Reply:

We reworded to 'Lead-based paint presents a similar concern.'

L242 This sentence is poorly worded and should be changed. Change 'is consequently resulting in' to 'results in an increasingly...'

Reply:

We reworded this sentence to '.... result in an increasingly diversified pathways of exposure.'

L243 'This' is an unspecific way to begin, you could connect to previous sentence with a ;

Reply:

We have removed the sentence for clarity.

L246 How are people ingesting near to lead sources? Needs explanation.

Reply: We have added the explanations. We now write

'...probably due to greater deposition onto food and subsequent consumption.'

L293 I am not sure 'outcomes' is the right word here. Also, wouldn't the effects you go on to list actually be cognitive.

Reply:

Thank you for pushing us on wording. We think of behavioural issues and criminal justice encounters, which are the focus of this paragraph, as non-cognitive outcomes

of lead-exposed children. However, to streamline the draft, we have reworded the paragraph to take out this labelling and merely discuss the evidence in the literature.

L326 Should this be 'confounding factors' or similar?

Yes, that is correct. We have re worded

Reviewer #2:

The authors have responded very effectively to the concerns raised in the previous reviews of this manuscript. The review is much more focused and better organized, and the writing is much tighter throughout.

Reply:

We sincerely thank the reviewer for the positive feedback. We greatly appreciate your thoughtful comments throughout the review process, which have helped us improve the clarity, focus, and overall quality of the manuscript.